# Inter-Device Reliability of a Three-Dimensional Markerless Motion Capture System Quantifying Elementary Movement Patterns in Humans

**DOI:** 10.3390/jfmk8020069

**Published:** 2023-05-22

**Authors:** Nicolas M. Philipp, Dimitrije Cabarkapa, Damjana V. Cabarkapa, Drake A. Eserhaut, Andrew C. Fry

**Affiliations:** Jayhawk Athletic Performance Laboratory—Wu Tsai Human Performance Alliance, Department of Health, Sport and Exercise Science, Lawrence, KS 66045, USA; d927c184@ku.edu (D.V.C.); drake.eserhaut@ku.edu (D.A.E.); acfry@ku.edu (A.C.F.)

**Keywords:** motion capture, biomechanics, markerless, movement, kinetic, kinematic, analysis

## Abstract

With advancements in technology able to quantify wide-ranging features of human movement, the aim of the present study was to investigate the inter-device technological reliability of a three-dimensional markerless motion capture system (3D-MCS), quantifying different movement tasks. A total of 20 healthy individuals performed a test battery consisting of 29 different movements, from which 214 different metrics were derived. Two 3D-MCS located in close proximity were utilized to quantify movement characteristics. Independent sample *t*-tests with selected reliability statistics (i.e., intraclass correlation coefficient (ICC), effect sizes, and mean absolute differences) were used to evaluate the agreement between the two systems. The study results suggested that 95.7% of all metrics analyzed revealed negligible or small between-device effect sizes. Further, 91.6% of all metrics analyzed showed moderate or better agreement when looking at the ICC values, while 32.2% of all metrics showed excellent agreement. For metrics measuring joint angles (198 metrics), the mean difference between systems was 2.9 degrees, while for metrics investigating distance measures (16 metrics; e.g., center of mass depth), the mean difference between systems was 0.62 cm. Caution is advised when trying to generalize the study findings beyond the specific technology and software used in this investigation. Given the technological reliability reported in this study, as well as the logistical and time-related limitations associated with marker-based motion capture systems, it may be suggested that 3D-MCS present practitioners with an opportunity to reliably and efficiently measure the movement characteristics of patients and athletes. This has implications for monitoring the health/performance of a broad range of populations.

## 1. Introduction

In recent years, the world of sport and human movement has experienced a great increase in the use and availability of technological devices to quantify and analyze various features of human health and performance. Amongst those technologies are biomechanical devices such as force platforms, quantifying movement characteristics from a kinetic standpoint, as well as motion capture systems focusing primarily on the acquisition of kinematic data such as joint ranges of motion and segment alignment. As part of this recent increase in the availability of sports and human performance technologies, scientists, researchers, as well as bioengineers have continuously studied the ways in which such devices may be created to collect data in a more time-efficient and user-friendly manner. For instance, once strictly laboratory-based force platforms are now portable and allow practitioners to gather data on large numbers of athletes or patients in minimal time. Similarly, advances in motion capture technology that have typically required the placement of reflective markers on various anatomical landmarks on the human body have resulted in markerless motion capture systems that do not require such markers on the human body and therefore may have a greater practical application in sport and health settings [1,2]. Beyond that, previous research reports have reported that the location and movement of the skin on which markers are placed, relative to actual skeletal motion and location, may present challenges with regard to the acquisition of repeatable and valid data in humans [3,4]. Within marker-based motion capture, this phenomenon is termed a soft tissue artifact and can lead to inaccuracies in the estimation of rigid body poses or kinematics [5].

Largely due to the above-mentioned limitations of marker-based systems, markerless motion capture solutions are starting to be explored in clinical and rehabilitation settings, as well as athletic settings [6]. A recently published study suggested that functional movement screening scores, non-invasively derived from a three-dimensional markerless motion capture system (3D-MCS), may provide health and fitness practitioners with key insights into a range of physical fitness parameters [7]. Particular markerless motion capture systems even advertise their capability of simultaneously quantifying kinetic data such as ground reaction forces, in addition to kinematic data. In 2016, Fry et al. [8] reported that ground reaction forces may be accurately derived from a motion capture system, using inverse dynamics, without the use of force platforms. Similarly, Cabarkapa et al. [9,10] suggested the reliability of utilizing markerless motion capture technology for measuring the kinetic properties of the basketball dunk, as well as the repeatability of motion health screening scores derived from an identical system. Being able to glean kinetic as well as kinematic data from only one system without having to apply reflective markers, or the additional use of force platforms would allow practitioners to test individuals in a more time-efficient manner. Within a recent SWOT (i.e., strength, weakness, opportunity, and threat) analysis looking at portable and low-cost markerless motion capture systems, Armitano-Lago et al. [2] proposed that markerless motion capture systems show considerable promise with regard to enhancing our understanding of human movement characteristics, especially in providing unrestricted and simple movement assessments in natural sporting contexts. While still limited, a growing body of literature has proposed the validity of markerless motion capture systems when compared to marker-based systems [11,12,13,14,15]. For instance, Sandau et al. [11] suggested that a markerless motion capture system was able to reliably produce data within the sagittal and frontal plane of motion during walking (e.g., joint flexion, extension, abduction, and adduction). However, data in the transverse plane (e.g., internal rotation, external rotations, eversions, and inversions) were deemed to be less reliable compared to those of a marker-based motion capture system. Looking at sagittal plane kinematics in a vertical jump task, Drazan et al. [12] found a very strong agreement between a custom markerless model approach and a gold-standard marker-based system. Further, Schmitz et al. [15] reported small differences in accuracy and reliability between a marker-based system and a single-camera markerless motion capture system. On the other hand, Harsted et al. [16] found reliability scores that were moderately acceptable for most measures but unacceptable for knee valgus and varus when comparing a markerless motion capture system to a traditional marker-based system during jumping tasks in preschool children. Similarly, Hando et al. [17] used a markerless motion capture system to identify potential associations between movement screening composite scores of vulnerabilities and injury risk in military trainees. In this study, the markerless motion capture composite scores only displayed poor to moderate test–retest reliability and failed to demonstrate the ability to discriminate between individuals that did and did not suffer subsequent musculoskeletal injuries [17]. It should be noted that the injuries were not classified as contact or non-contact injuries, which makes interpretation of the data challenging [17]. Still in their infancy, other works in the existing literature within the field of health and sport have used markerless motion capture systems to gain insights into human health and movement characteristics [10,18,19,20]. While the previously highlighted literature suggests the potentially effective use of markerless motion capture technologies, particularly from a validity standpoint, certain degrees of uncertainty pertaining to the reliability of such devices still remain, especially across a wide range of movement tasks and variables. Additionally, the reliability of markerless motion capture includes both technological reliability (e.g., between-device agreement), as well as biological reliability, testing the ability of a human to adequately repeat the motions being tested.

With the previously highlighted evolution of innovative markerless motion capture systems in mind, the aim of the present study was to determine the inter-device reliability (i.e., technical reliability) between two identical markerless motion capture systems placed in close proximity to each other. We hypothesized that the two identical systems would reflect good technological reliability for a variety of different movement tasks and the respective joint and segmental angle variables and center of mass distance measures in healthy individuals. The authors see value in investigating the novel technologies’ reliability (i.e., technical and biological) prior to comparison with established industry gold standards for validity purposes.

## 2. Materials and Methods

### 2.1. Participants

A total of 20 healthy men (*n* = 11, height = 181 ± 7.2 cm, body mass = 87.7 ± 11.1 kg, age = 26.8 ± 6.8 years) and women (*n* = 9, height = 167 ± 6.6 cm, body mass = 62.7 ± 6.9 kg, age = 24.2 ± 7.3 years) volunteered to participate in the study. Prior to any testing, the subjects completed a health history questionnaire, indicating they were free of musculoskeletal injury. All participants signed an informed consent form. All respective study procedures were approved by the University of Kansas’s Institutional Review Board.

### 2.2. Procedures

As part of this study, all subjects performed a total of 29 different movements (Table 1). These procedures were preceded by a dynamic warm-up protocol that included cycling on a stationary bike for 5 min. Relevant kinematic data from these movements were quantified using two 3D-MCS (DARI Motion, Overland Park, KS, USA) composed of eight high-definition cameras recording at 60 fps. These cameras were attached to a metal frame surrounding the testing area. Corresponding cameras from each system were placed next to each other in close proximity. General data collection procedures for this study were adapted from Cabarkapa et al. (2022). The hull technology model records and subtracts the visual signal minus the background, which is used to generate a pixelated person in order to obtain biomechanical parameters of interest. Following manufacturer guidelines, each system was separately calibrated prior to testing. Specific movement tasks were explained and demonstrated by the principal investigator of the study. Following this demonstration, the member of the research team running the motion capture system provided the subject with the following command: “three, two, one, go”. Following the “go” command, the subject completed the movement task which was being recorded by the two 3D-MCSs. After the completion of the respective movement task, the command “done” was provided to the subject, to indicate the end of the movement. Instructions for the completion of all 29 movements remained identical for all 20 subjects. A total of 214 metrics were derived from the movement battery, including 198 joint and segment angle variables, and 16 distance measures (e.g., center of mass movement).

### 2.3. Statistical Analysis

Prior to any analyses, all data were checked for normal distribution using Shapiro–Wilk’s statistics. To determine between-device differences (System 1 vs. System 2), independent *t*-tests were used for all variables of interest. Data with a normal distribution were analyzed using Student’s independent *t*-test, while the Mann–Whitney U statistical test was used for data that were not normally distributed. For the student’s *t*-tests, mean and standard deviation values were reported, while the median was reported for the Mann–Whitney U tests. For parametric data, Cohen’s d effect sizes were calculated and interpreted as negligible (≤0.10), small (0.11–0.50), moderate (0.51–0.75), and large (>0.75) [21]. Effect sizes for non-parametric data were interpreted as described within the previous sentence, following a conversion from η2 to Cohen’s d [22,23]. Additionally, intraclass correlation coefficients (ICC) were used to examine the agreement between respective metrics of interest. ICCs were interpreted following suggestions by Koo and Li [24], where <0.50 was deemed poor reliability, 0.50–0.74 was deemed moderate reliability, 0.75–0.90 was deemed good reliability, and >0.90 was deemed excellent reliability. Lastly, mean absolute between-system differences were reported to indicate the actual difference between systems for distance measures (cm) and degrees (deg). All statistical inferences were made using an alpha level of <0.05. Data were analyzed using the R statistical computing environment and language (v. 4.0; R Core Team, 2020) via the Jamovi graphical user interface.

## 3. Results

Descriptive statistics for all variable comparisons may be found in the supplementary file (Appendix A). Of the 214 variables reported, 94.9% of the metrics revealed negligible or small between-device effect sizes. Further, 91.6% of all metrics analyzed showed moderate or better agreement when looking at the ICC values, including 32.2% of all metrics showing excellent agreement. Only 2.3% of all metrics were significantly different when compared to the other system. Summary statistics for effect sizes and ICC values may be found in Table 2 and Table 3, respectively. For metrics measuring joint angles, the mean absolute difference between systems was 2.9 degrees (198 metrics), while for metrics investigating distance measures (e.g., center of mass depth), the mean difference between systems was 0.62 cm (16 metrics).

## 4. Discussion

While previous research reports have investigated the biological reliability and validity of variables derived from 3D-MCSs [10,11,17], this study aimed to quantify technological reliability for the underlying variables from which all calculated measures are derived. More specifically, the aim of this study was to quantify the between-device agreement of two identical markerless motion capture systems located in close proximity. To the authors knowledge, this is the first study investigating the inter-device reliability of a markerless motion-capture system, capturing a plethora of elementary movements, from which a wide range of metrics are gleaned. Study findings revealed that a broad range of reliability scores (i.e., ICC or ES) were found across the selected metrics. Up to 75% of the metrics presented ICC scores of 0.70 or higher, reflecting moderate to excellent agreement. Similarly, when looking at effect sizes for between-system comparisons, 95.7% of all metrics suggested small or negligible effect sizes. Previous research reports looking at the reliability of markerless motion capture systems reported good reliability for movements performed within the sagittal or frontal plane, while movements performed within the transverse plane (e.g., rotations) revealed fewer stable measures [11]. Within our data, this suggestion is only partially reflected. When looking at internal and external shoulder rotation, all four metrics present ICC values of 0.90 or higher. For the trunk rotation exercise, ICC values for lumbar and thoracic rotation range from 0.69–0.81. Lastly, for the reverse lunge with rotation movement, ICC values for lumbar and thoracic rotation only range from 0.29 to 0.78. However, the metric presenting the 0.29 ICC value was accompanied by a small effect size. Our data suggest that reliability may not only be influenced by the plane in which movements are performed but also by the specific movements and body parts that are being investigated. Hip angles displayed mean absolute differences ranging from 0.3 degrees to 11.4 degrees, with the average being 6.3 degrees across 28 different hip flexion measures. While looking at the validity of a markerless motion capture system, Harsted et al. [16] reported moderate to poor agreement for a range of hip flexion measures extracted from squats and jumps, when comparing the markerless motion capture system to a marker-based system. In their study, between-system differences ranged from 5.8 degrees to 14.8 degrees [16].

A very commonly implemented and analyzed movement from a rehabilitation and athletic performance standpoint is the countermovement vertical jump [12]. While using a different analysis technique, Drazan et al. [12] found very strong agreement between a markerless motion capture technology and a gold-standard marker-based system when looking at a number of different angular measurements of the hip, knee, and ankle, for the vertical jump. In our study, moderate to excellent agreement between the two markerless motion capture systems were found for all countermovement vertical jump metrics, except for ankle flexion and knee valgus during the eccentric phase of the jump, and upon landing. Similarly, for the drop jump, ICC values for ankle flexion and dynamic valgus during landing only ranged from 0.29 to 0.49, with effect sizes of 0.10 to 0.62. All kinematic measures within the countermovement vertical and drop jump presented considerably more between-system agreement. It should be noted that many of the valgus measures were made during activities not typically performed for clinical assessments and must be interpreted accordingly. When looking at the upper body movements, 9 out of 12 metrics presented ICC values of good to excellent agreement. One should be cognizant that the shoulder flexion metrics presenting with lower ICC values showed non-significant differences, and small to negligible effect sizes, suggesting small to negligible between-system differences. Distance measures such as the center of mass depth during different movement tasks such as vertical jumps have been of interest to practitioners working in sport performance settings. For instance, Merrigan et al. [25] suggested that the center of mass depth during a countermovement vertical jump was an important contributor to overall jumping capability. Within our data, distance measures such as center of mass depths were in good agreement between the two systems, indicated by only two metrics presenting ICC values under 0.90, and only one metric presenting an ICC value under 0.85. In dynamic movements such as countermovement vertical jumps, the center of mass moves through a significant range of motion, making it important to acquire a reliable measure of center of mass movements. Within marker-based systems, markers attached to the skin using double adhesive tape can influence normal movement patterns and can move relative to the underlying bone, commonly known as a skin movement artifact or soft tissue artifact [3,4,5]. Beyond that, certain movements, clothes worn, or ranges of motion can even lead to further issues with regard to these reflective markers. A recent study even had to standardize the squat depth of the participants in order to avoid the occlusion of the reflective markers on the anterior superior illicit spine at lower squat depths [26]. This could certainly restrict an individual’s normal range of motion, posing limitations to the acquisition of practically relevant data. Further, this may influence not only the efficiency of assessment procedures but also potentially the reliability of derived metrics.

Readers should be cognizant of the potential limitations when interpreting the results of this study. From a procedural perspective, given the broad range of movement tasks, participants only performed one repetition for each movement, which is in line with the suggestions for test administration given by the manufacturer of the motion capture system. Future studies may aim to narrow in on the specific movements identified within this study, investigating aspects of reliability across additional repetitions. Pending appropriate reliability studies, technologies such as the one used in this study should further be compared to marker-based tracking systems to gain added insights into the validity of markerless motion capture systems. Additionally, the breadth and depth of the movements and respective metrics did not allow the authors to provide a detailed discussion of all variables. Furthermore, the landscape of available markerless motion capture solutions is rapidly expanding, making it difficult to generalize findings from this study across other systems and software. Lastly, the group of participants consisted entirely of healthy individuals. Assuming that healthy individuals present less variability when performing the movement battery, future investigations may aim to replicate study procedures within injured or rehabilitating groups. This may lend insights into the between-device reliability of markerless motion capture systems across a broader range of movement characteristics. Especially in return-to-play scenarios for athletes a more frequent evaluation of movement characteristics could greatly aid clinical practitioners in evaluating patients’ progress as well as in their ability to individually tailor a specific return-to-play protocol [18]. In many cases, this assessment frequency is hindered by the cumbersome and time-consuming nature of marker-based motion capture systems. Therefore, 3D-MCSs could greatly enhance the work of a broad range of practitioners as well as the health and performance of different groups of individuals.

## 5. Conclusions

The study findings suggested promising results with regard to the between-device reliability of a 3D-MCS quantifying a broad range of movements. Overall, nearly all of the variables assessed demonstrated acceptable to strong inter-device reliability, indicating low technological variability. Readers are encouraged to employ caution when trying to generalize the results of this study past the specific system and software used. Given the technological reliability reported in this study as well as the logistical and time-related limitations associated with marker-based motion capture systems, it is concluded that this specific markerless motion capture technology presents practitioners with an opportunity to measure the movement characteristics of patients and athletes at a greater frequency, due in part to its high technological reliability. This could have implications for the health and performance of a broad range of populations.

## Figures and Tables

**Table 1 jfmk-08-00069-t001:** List and description of all 29 tested movements.

Specific Movement Performed	Description of Movement
Shoulder Abduction	Start with arms at your sides with your palms facing forward. With arms straight, raise them out from your sides and over your head (abduct), keeping palms forward throughout the entire movement
Shoulder Horizontal Abduction	Start with your arms out in front of you at shoulder height with your palms facing each other. Bring your arms away from each other and behind your body as far as possible, keeping them at shoulder height throughout
Shoulder Internal/External Rotation	Start with elbows and shoulders bent at 90 degrees and palms facing down. Rotate arms up and back as far as possible (externally), and then forward and down (internally). Keeping elbows in the same spot during the movement
Shoulder Flexion/Extension	Begin with arms by your side. In one fluid motion, bring hands forward and up above the head, then down and back behind the body, and then return to original position.
Forward Fold	Begin with feet shoulder width apart. Tuck chin to chest and continue to round the back forward, bending at the hips in an attempt to touch the forehead to the knees.
Trunk Lateral Flexion Right	Begin with feet shoulder width apart and hands by the sides. Keep right hand on the outside of the right leg and bend upper body to the right as far down as possible, then return to starting position
Trunk Lateral Flexion Left	Begin with feet shoulder width apart and hands by the sides. Keep left hand on the outside of the right leg and bend upper body to the right as far down as possible, then return to starting position
Trunk Rotation	Start with elbows and shoulders bent at 90 degrees and palms facing down. In one fluid motion, rotate arms, torso, and head, first to the right, then to the left, and then return to starting position
Reverse Lunge with Rotation Right	Begin with arms out to the sides and elbows bent. Reach left leg back and drop into lunge without letting left knee touch the ground. At the bottom of the lunge, rotate the trunk to the right as far as possible, then return to starting position
Reverse Lunge with Rotation Left	Begin with arms out to the sides and elbows bent. Reach right leg back and drop into lunge without letting right knee touch the ground. At the bottom of the lunge, rotate the trunk to the left as far as possible, then return to starting position
Body Weight Squat	Begin with feet shoulder width apart and toes pointing forward. In one fluid motion, squat as low as possible, then return to the starting position
Overhead Squat	Begin with feet shoulder width apart, toes pointing forward and the dowel rod held above the head, with hands positioned wider than shoulders. In one fluid motion, squat as low as possible, and return to the original position
Forward Lunge Right	Begin by striding out with right leg getting as far and deep as possible. Then return to the starting position in one fluid motion. During movement keep arms out for balance
Forward Lunge Left	Begin by striding out with left leg getting as far and deep as possible. Then return to the starting position in one fluid motion. During movement keep arms out for balance
Lateral Lunge Right	Begin by stepping out as far to the right as possible. While allowing arms to travel out in front of the body, lunge as low as possible. Then return to the starting position
Lateral Lunge Left	Begin by stepping out as far to the left as possible. While allowing arms to travel out in front of the body, lunge as low as possible. Then return to the starting position
Standing Hip Abduction Right	Begin with hands on hips, standing with feet together. Keep right leg straight and raise it out to the side as far as possible, then return to the starting position
Standing Hip Abduction Left	Begin with hands on hips, standing with feet together. Keep left leg straight and raise it out to the side as far as possible, then return to the starting position
Unilateral Squat Right	Transfer weight to the right leg, lifting the left foot off the ground and behind the body. In one fluid motion, squat as low as possible, keeping the left foot off the ground, and arms out for balance
Unilateral Squat Left	Transfer weight to the right leg, lifting the left foot off the ground and behind the body. In one fluid motion, squat as low as possible, keeping the left foot off the ground, and arms out for balance
Countermovement Vertical Jump	Begin by standing with feet shoulder width apart. Load and jump as high as possible. Do not step into the jump, but you may use an arm swing
Static Vertical Jump	Begin by standing with feet shoulder width apart. Lower into a squat position with arms repositioned to a natural jumping stance. Remain in this position for two seconds. On the signal “jump” immediately jump as high as possible from the squat position
Unilateral Vertical Jump Right	Begin by standing on right leg with left foot off the ground behind the body. Load and jump as high as possible, using an arm swing, and landing on your right foot again
Unilateral Vertical Jump Left	Begin by standing on left leg with left foot off the ground behind the body. Load and jump as high as possible, using an arm swing, and landing on your left foot again
Lateral Bound Right	Begin by taking two large steps to the left. Push off with the left leg and bound as far to the right side as possible. Land on the right leg and immediately push off in the opposite direction to reach the starting position
Lateral Bound Left	Begin by taking two large steps to the right. Push off with the right leg and bound as far to the left side as possible. Land on the left leg and immediately push off in the opposite direction to reach the starting position
5 Hop Right	Begin standing on the right leg with left foot off the ground behind the body. Jump on the right leg five times. Jump as high as possible, and as fast as possible, spending as little time on the ground between jumps as possible
5 Hop Left	Begin standing on the left leg with right foot off the ground behind the body. Jump on the left leg five times. Jump as high as possible, and as fast as possible, spending as little time on the ground between jumps as possible
Drop Jump	Begin standing on a 30-cm-high box. With either foot, step off the box landing on two feet. Immediately jump for maximal height, spending as little time as possible on the ground. An arm swing may be used

**Table 2 jfmk-08-00069-t002:** Summary statistics for effect sizes.

Effect Size	Number of Variables (% From Total)
Negligible (≤0.10)	72 variables (33.7%)
Small (0.11–0.50)	131 variables (61.2%)
Moderate (0.51–0.75)	11 variables (5.1%)
Large (>0.75)	0 variables (0.0%)

**Table 3 jfmk-08-00069-t003:** Summary statistics for interclass correlations coefficients (ICC) values.

ICC	Metric Count (*n* = 214)	Total	Metric Count (% of Total)	Total (%)
≥0.90	69	-	32.2%	-
≥0.80–0.89	54	123	25.3%	57.5%
≥0.70–0.79	38	161	17.8%	75.2%
≥0.60–0.69	19	180	8.9%	84.1%
≥0.50–0.59	16	196	7.5%	91.6%
≥0.40–0.49	6	202	2.8%	94.4%
≥0.30–0.39	9	211	4.2%	98.6%
≥0.20–0.29	3	214	1.4%	100%
<0.20	0	-	-	-

## Data Availability

The data presented in this study are available upon request from the corresponding author.

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
