# Peer review of "Inter-Device Reliability of a Three-Dimensional Markerless Motion Capture System Quantifying Elementary Movement Patterns in Humans"

_jfmk, 2023, doi:10.3390/jfmk8020069_

Round 1

Reviewer 1 Report

This is a well constructed report. You are to be congratulated!!! 

The only two very minor comments I would make are:

your population is young healthy individuals. As we know either through disease process or aging movement patterns alter, This is not reflected in the submission.

The battery of tests is comprehensive but not reflective of what participants can undertake.

well done

Danny

Author Response

Dear Reviewer, thank you for a critical review of our work. We believe that based on your comments, our work is improved and better suited for readership.

your population is young healthy individuals. As we know either through disease process or aging movement patterns alter

 This is not reflected in the submission.

  • "Dear reviewer, thank you for this comment. We agree with you and have added an additional sentence to our discussion highlighting this as a limitation of the study, and suggesting that future research should aim to replicate methodologies across a wider range of individuals."

The battery of tests is comprehensive but not reflective of what participants can undertake.

  • "Dear reviewer, we agree with you. However, the technology used within our investigation has a pre-set selection of exercises. Future research should aim to expand on the test batteries reflective of what participants can undertake."

We hope by addressing the above comments, we suffice what you were looking for.

Reviewer 2 Report

This clearly written manuscript investigates the inter-device reliability of two identical, marker-less motion capture systems. The manuscript is logically organized and appropriate in length, the data collection and data analysis are rigorous and appropriate. My main questions relate to why the authors did not also compare results of the marker-less systems to results from systems using markers or for example moving fluoroscopy systems which would be the gold standard? It could be made more clear what the goal is of comparing two systems that are identical while not exactly knowing how valid the systems are?

Title

The title is informative and clear.

Abstract and keywords

The abstract is coherent, factual, and clearly summarizes the manuscript.

Introduction

Line 117-120: what is meant here? This is rather unclear. Because your publication also does not present any raw data?

Methods

Methods are clearly written and logically organized.

Results

Results are well organized. It would be valuable to show some examples of raw data, for example joint angles over time and compare between the systems.

Discussion

Lines 219-222: can you add more detailed description of how exactly your study is different from the other ones and discuss what they have done?

Author Response

Dear Reviewer, thank you for a critical review of our work. We believe that based on your comments, our work is improved and better suited for readership.

This clearly written manuscript investigates the inter-device reliability of two identical, marker-less motion capture systems. The manuscript is logically organized and appropriate in length, the data collection and data analysis are rigorous and appropriate. My main questions relate to why the authors did not also compare results of the marker-less systems to results from systems using markers or for example moving fluoroscopy systems which would be the gold standard? It could be made more clear what the goal is of comparing two systems that are identical while not exactly knowing how valid the systems are?

  • "Dear reviewer, thank you for your comment. We hope the following explanation will clarify your questions about the scope of the study. While it would certainly be interesting and have clear merit to conduct further studies comparing markerless systems to what is considered the gold standard in marker-based systems, this was not the aim of the study. The primary aim of the study was to investigate the technological / between-device reliability of the markerless system used. While the validity piece is obviously an important step in investigating the potential usefulness of markerless motion capture systems, we believe that investigations into the technological reliability are important with regards to research from the reliability perspective. Once reliability is given, our goal in future projects is to validate markerless motion capture against the industry gold standard which are marker-based systems."

Introduction

Line 117-120: what is meant here? This is rather unclear. Because your publication also does not present any raw data?

  • "Dear reviewer, thank you for your comment. We agree and have eliminated the respective sentence from the introduction."

Results

Results are well organized. It would be valuable to show some examples of raw data, for example joint angles over time and compare between the systems.

  • "Dear reviewer, thank you for a careful review of our work. We agree that it would be valuable to show some examples of the raw data, however, the technology used made it rather cumbersome for us to visualized the raw data. If this is required for publication, we'd be open to reaching our to the company to look into other options for visualizing raw data."

Discussion

Lines 219-222: can you add more detailed description of how exactly your study is different from the other ones and discuss what they have done?

- "Dear reviewer, thank you for this comment. We have added the following sentence to the specified location in the discussion: “More specifically, the aim of the study was to quantify the between-device agreement of two identical markerless motion-capture systems located in close proximity.” We hope this comment suffices what you were asking for." Furthermore, we have added additional sentences to the introduction to clearly state the purpose of this investigation, which was to look into the technological reliability (between-device agreement), rather than validity or biological reliability. 

We hope the above responses suffice what you were asking for. 

Reviewer 3 Report

The authors performed a study to test the inter-device reliability of a three-dimensional markerless motion capture system. This is an interesting topic; however, the most important thing that users care about is the performance of the markerless system. Can the performance of a markerless system compete with the marker tracking system? For example, the following 3 papers compared the marker-based and markerless methods.

1.      Markerless motion capture: What clinician-scientists need to know right now, JSAMS Plus, Volume 1, October 2022, 100001. (https://doi.org/10.1016/j.jsampl.2022.100001)

2.      Accuracy of a markerless motion capture system in estimating upper extremity kinematics during boxing, Front. Sports Act. Living, 25 July 2022. (https://doi.org/10.3389/fspor.2022.939980)

3.      Comparison of Markerless and Marker-Based Motion Capture Technologies through Simultaneous Data Collection during Gait: Proof of Concept, PLOS ONE, 2014. (https://doi.org/10.1371/journal.pone.0087640)

In the experimental design, the data measured by a new method (markerless method) should be used to compare the ground truth or the golden standard (marker-based method). The study presents some promising aspects, but there is still scope for enhancing its experimental design and creativity. Other comments are listed as follows.

1)      The abstract is too long. The authors need to provide a concise abstract with word numbers ranging from 200 to 250 words.

2)      The full name of ICC should be provided in the abstract.

3)      While mentioning a literature, the citation mark should be added immediately. In line 74, (2020, 2022) needs to be added after “Cabarkapa et al.”. The authors need to check throughout the manuscript.

4)      Table 2 should be renamed as Table S1 if the authors want to place this Table in a supplementary file.

Author Response

Dear Reviewer, thank you for a critical review of our work. We believe that based on your comments, our work is improved and better suited for readership. Changes and additions to the updated manuscript are highlighted in blue. 

The authors performed a study to test the inter-device reliability of a three-dimensional markerless motion capture system. This is an interesting topic; however, the most important thing that users care about is the performance of the markerless system. Can the performance of a markerless system compete with the marker tracking system? For example, the following 3 papers compared the marker-based and markerless methods.

  • "Dear reviewer, thank you for a careful review of our work. We hope the following explanation will clarify your questions about the overarching scope of the study. While it would certainly be interesting and have clear merit to conduct further studies comparing markerless systems to what is considered the gold standard in marker-based systems, this was not the aim of the study. The primary aim of the study was to investigate the technological / between-device reliability of the markerless system used. While the validity piece is obviously an important step in investigating the potential usefulness of markerless motion capture systems, we believe that investigations into the technological reliability are important with regards to research from the reliability perspective. Pending appropriate reliability studies, our goal in future projects is to further validate markerless motion capture against the industry gold standard which are marker-based systems. We have added numerous statements to the manuscript (highlighted in blue) clarifying the aim of the study"

1)      The abstract is too long. The authors need to provide a concise abstract with word numbers ranging from 200 to 250 words.

  • "Dear reviewer, thank you for pointing this out to us. We have significantly shortened the abstract to now be 254 words long instead of 300 words. We hope this suffices what you were asking for."

2)      The full name of ICC should be provided in the abstract.

  • "Dear reviewer, thank you for this comment. To our understanding, the full name of ICC is currently provided within the abstract (line 19). We have added the abbreviation “ICC” in parathesis, after the first time we have mentioned the full name."

3)      While mentioning a literature, the citation mark should be added immediately. In line 74, (2020, 2022) needs to be added after “Cabarkapa et al.”. The authors need to check throughout the manuscript.

  • "Dear reviewer, thank you for pointing this out to us. We have made the respective change within our manuscript."

4)      Table 2 should be renamed as Table S1 if the authors want to place this Table in a supplementary file.

- "Dear reviewer, thank you for a careful revision of our work. We renamed table 2 as table S1. We hope this suffices what you were asking for."

Round 2

Reviewer 2 Report

The authors carefully responded to each of the reviewers' comments and clarity of the manuscript further improved. If possible to include raw data this would be highly recommended and further improve the quality of the manuscript. 

Author Response

Dear reviewer, thank you for your comments, and a clear review of our work. Again, we believe that based on your comments, our work is improved, and better suited for readership.

The authors carefully responded to each of the reviewers' comments and clarity of the manuscript further improved. If possible to include raw data this would be highly recommended and further improve the quality of the manuscript. 

  • Dear reviewer, thank you for this comment. While we understand your recommendation of including raw data, after discussion with the Lab Director, we came to the common conclusion as a team that we will not be able to provide the raw data. We believe the breath and depth of the data presented in our manuscript is sufficient for readership.

Thank you for your help in this process. 

Reviewer 3 Report

The authors answered questions appropriately. The format of Tables 3 and 4 needs to adjust to fulfill the specified table format of this journal. The font in Table 1 is not consistent with other parts of the manuscript. Please check and revise them.

Author Response

Dear reviewer, thank you for your comments, and a detailed review of our work. Again, we believe that based on your comments, our work is improved, and better suited for readership. See below our responses to you recent comments. 

The authors answered questions appropriately. The format of Tables 3 and 4 needs to adjust to fulfill the specified table format of this journal. The font in Table 1 is not consistent with other parts of the manuscript. Please check and revise them.

  • Dear reviewer, thank you for your comments. In our updated manuscript, we have updated the formats of tables 3 and 4. We hope this suffices what you were asking for. Further, we have changed the font in table 1 to match the font throughout the manuscript.

Again, thank you for your help in this process.